# Predictive performance of multi-model ensemble forecasts of COVID-19 across European nations

Katharine Sherratt[1]*, Hugo Gruson[1], Rok Grah[2], Helen Johnson[2], Rene Niehus[2], Bastian Prasse[2], Frank Sandmann[2], Jannik Deuschel[3], Daniel Wolffram[3], Sam Abbott[1], Alexander Ullrich[4], Graham Gibson[5], Evan L Ray[5], Nicholas G Reich[5], Daniel Sheldon[5], Yijin Wang[5], Nutcha Wattanachit[5], Lijing Wang[6], Jan Trnka[7], Guillaume Obozinski[8], Tao Sun[8], Dorina Thanou[8], Loic Pottier[9], Ekaterina Krymova[10], Jan H Meinke[11], Maria Vittoria Barbarossa[12], Neele Leithauser[13], Jan Mohring[13], Johanna Schneider[13], Jaroslaw Wlazlo[13], Jan Fuhrmann[14], Berit Lange[15], Isti Rodiah[15], Prasith Baccam[16], Heidi Gurung[16], Steven Stage[17], Bradley Suchoski[16], Jozef Budzinski[18], Robert Walraven[19], Inmaculada Villanueva[20], Vit Tucek[21], Martin Smid[22], Milan Zajicek[22], Cesar Perez Alvarez[23], Borja Reina[23], Nikos I Bosse[1], Sophie R Meakin[1], Lauren Castro[24], Geoffrey Fairchild[24], Isaac Michaud[24], Dave Osthus[24], Pierfrancesco Alaimo Di Loro[25], Antonello Maruotti[25], Veronika Eclerova[26], Andrea Kraus[26], David Kraus[26], Lenka Pribylova[26], Bertsimas Dimitris[27], Michael Lingzhi Li[27], Soni Saksham[27], Jonas Dehning[28], Sebastian Mohr[28], Viola Priesemann[28], Grzegorz Redlarski[29], Benjamin Bejar[30], Giovanni Ardenghi[31], Nicola Parolini[31], Giovanni Ziarelli[31], Wolfgang Bock[32], Stefan Heyder[33], Thomas Hotz[33], David E Singh[34], Miguel Guzman-Merino[34], Jose L Aznarte[35], David Morina[36], Sergio Alonso[37], Enric Alvarez[37], Daniel Lopez[37], Clara Prats[37], Jan Pablo Burgard[38], Arne Rodloff[39], Tom Zimmermann[39], Alexander Kuhlmann[40], Janez Zibert[41], Fulvia Pennoni[42], Fabio Divino[43], Marti Catala[44], Gianfranco Lovison[45], Paolo Giudici[46], Barbara Tarantino[46], Francesco Bartolucci[47], Giovanna Jona Lasinio[48], Marco Mingione[48], Alessio Farcomeni[49], Ajitesh Srivastava[50], Pablo Montero-Manso[51], Aniruddha Adiga[52], Benjamin Hurt[52], Bryan Lewis[52], Madhav Marathe[52], Przemyslaw Porebski[52], Srinivasan Venkatramanan[52], Rafal P Bartczuk[53], Filip Dreger[53], Anna Gambin[53], Krzysztof Gogolewski[53], Magdalena Gruziel-Slomka[53], Bartosz Krupa[53], Antoni Moszyński[53], Karol Niedzielewski[53], Jedrzej Nowosielski[53], Maciej Radwan[53], Franciszek Rakowski[53], Marcin Semeniuk[53], Ewa Szczurek[53], Jakub Zielinski[53], Jan Kisielewski[53,54], Barbara Pabjan[55], Kirsten Holger[56], Yuri Kheifetz[56], Markus Scholz[56], Biecek Przemyslaw[57]†, Marcin Bodych[58], Maciej Filinski[58], Radoslaw Idzikowski[58], Tyll Krueger[58], Tomasz Ozanski[58], Johannes Bracher[3], Sebastian Funk[1]

*For correspondence:
katharine.sherratt@lshtm.ac.uk

Present address: †University of Warsaw, Warsaw, Poland

[1]London School of Hygiene & Tropical Medicine, London, United Kingdom; [2]European Centre for Disease Prevention and Control (ECDC), Stockholm, Sweden; [3]Karlsruhe Institute of Technology, Karlsruhe, Germany; [4]Robert Koch Institute, Berlin, Germany; [5]University of Massachusetts Amherst, Amherst, United States; [6]Boston Children's Hospital and Harvard Medical School, Boston, United States; [7]Third Faculty of Medicine, Charles University, Prague, Czech Republic; [8]Ecole Polytechnique Federale de Lausanne, Lausanne, Switzerland; [9]Éducation nationale, Valbonne, France; [10]Eidgenossische Technische Hochschule, Zurich, Switzerland; [11]Forschungszentrum

Jülich GmbH, Jülich, Germany; [12]Frankfurt Institute for Advanced Studies, Frankfurt, Germany; [13]Fraunhofer Institute for Industrial Mathematics, Kaiserslautern, Germany; [14]Heidelberg University, Heidelberg, Germany; [15]Helmholtz Centre for Infection Research, Braunschweig, Germany; [16]IEM, Inc, Bel Air, United States; [17]IEM, Inc, Baton Rouge, United States; [18]Independent researcher, Vienna, Austria; [19]Independent researcher, Davis, United States; [20]Institut d'Investigacions Biomèdiques August Pi i Sunyer, Universitat Pompeu Fabra, Barcelona, Spain; [21]Institute of Computer Science of the CAS, Prague, Czech Republic; [22]Institute of Information Theory and Automation of the CAS, Prague, Czech Republic; [23]Inverence, Madrid, Spain; [24]Los Alamos National Laboratory, Los Alamos, United States; [25]LUMSA University, Rome, Italy; [26]Masaryk University, Brno, Czech Republic; [27]Massachusetts Institute of Technology, Cambridge, United States; [28]Max-Planck-Institut für Dynamik und Selbstorganisation, Göttingen, Germany; [29]Medical University of Gdansk, Gdańsk, Poland; [30]Paul Scherrer Institute, Villigen, Switzerland; [31]Politecnico di Milano, Milan, Italy; [32]Technical University of Kaiserlautern, Kaiserslautern, Germany; [33]Technische Universität Ilmenau, Ilmenau, Germany; [34]Universidad Carlos III de Madrid, Leganes, Spain; [35]Universidad Nacional de Educación a Distancia (UNED), Madrid, Spain; [36]Universitat de Barcelona, Barcelona, Spain; [37]Universitat Politècnica de Catalunya, Barcelona, Spain; [38]Universitat Trier, Trier, Germany; [39]University of Cologne, Cologne, Germany; [40]University of Halle, Halle, Germany; [41]University of Ljubljana, Ljubljana, Slovenia; [42]University of Milano-Bicocca, Milano, Italy; [43]University of Molise, Pesche, Italy; [44]University of Oxford, Oxford, United Kingdom; [45]University of Palermo, Palermo, Italy; [46]University of Pavia, Pavia, Italy; [47]University of Perugia, Perugia, Italy; [48]University of Rome "La Sapienza", Rome, Italy; [49]University of Rome "Tor Vergata", Rome, Italy; [50]University of Southern California, Los Angeles, United States; [51]University of Sydney, Sydney, Australia; [52]University of Virginia, Charlottesville, United States; [53]University of Warsaw, Warsaw, Poland; [54]University of Bialystok, Warsaw, Poland; [55]University of Wroclaw, Wroclaw, Poland; [56]Universtät Leipzig, Leipzig, Germany; [57]Warsaw University of Technology, Warsaw, Poland; [58]Wroclaw University of Science and Technology, Wroclaw, Poland

## Abstract

**Background:** Short-term forecasts of infectious disease burden can contribute to situational awareness and aid capacity planning. Based on best practice in other fields and recent insights in infectious disease epidemiology, one can maximise the predictive performance of such forecasts if multiple models are combined into an ensemble. Here, we report on the performance of ensembles in predicting COVID-19 cases and deaths across Europe between 08 March 2021 and 07 March 2022.

**Methods:** We used open-source tools to develop a public European COVID-19 Forecast Hub. We invited groups globally to contribute weekly forecasts for COVID-19 cases and deaths reported by a standardised source for 32 countries over the next 1–4 weeks. Teams submitted forecasts from March 2021 using standardised quantiles of the predictive distribution. Each week we created an ensemble forecast, where each predictive quantile was calculated as the equally-weighted average (initially the mean and then from 26th July the median) of all individual models' predictive quantiles. We measured the performance of each model using the relative Weighted Interval Score (WIS), comparing models' forecast accuracy relative to all other models. We retrospectively explored alternative methods for ensemble forecasts, including weighted averages based on models' past predictive performance.

**Results:** Over 52 weeks, we collected forecasts from 48 unique models. We evaluated 29 models' forecast scores in comparison to the ensemble model. We found a weekly ensemble had a consistently strong performance across countries over time. Across all horizons and locations, the

ensemble performed better on relative WIS than 83% of participating models' forecasts of incident cases (with a total N=886 predictions from 23 unique models), and 91% of participating models' forecasts of deaths (N=763 predictions from 20 models). Across a 1–4 week time horizon, ensemble performance declined with longer forecast periods when forecasting cases, but remained stable over 4 weeks for incident death forecasts. In every forecast across 32 countries, the ensemble outperformed most contributing models when forecasting either cases or deaths, frequently outperforming all of its individual component models. Among several choices of ensemble methods we found that the most influential and best choice was to use a median average of models instead of using the mean, regardless of methods of weighting component forecast models.

**Conclusions:** Our results support the use of combining forecasts from individual models into an ensemble in order to improve predictive performance across epidemiological targets and populations during infectious disease epidemics. Our findings further suggest that median ensemble methods yield better predictive performance more than ones based on means. Our findings also highlight that forecast consumers should place more weight on incident death forecasts than incident case forecasts at forecast horizons greater than 2 weeks.

**Funding:** AA, BH, BL, LWa, MMa, PP, SV funded by National Institutes of Health (NIH) Grant 1R01GM109718, NSF BIG DATA Grant IIS-1633028, NSF Grant No.: OAC-1916805, NSF Expeditions in Computing Grant CCF-1918656, CCF-1917819, NSF RAPID CNS-2028004, NSF RAPID OAC-2027541, US Centers for Disease Control and Prevention 75D30119C05935, a grant from Google, University of Virginia Strategic Investment Fund award number SIF160, Defense Threat Reduction Agency (DTRA) under Contract No. HDTRA1-19-D-0007, and respectively Virginia Dept of Health Grant VDH-21-501-0141, VDH-21-501-0143, VDH-21-501-0147, VDH-21-501-0145, VDH-21-501-0146, VDH-21-501-0142, VDH-21-501-0148. AF, AMa, GL funded by SMIGE - Modelli statistici inferenziali per governare l'epidemia, FISR 2020-Covid-19 I Fase, FISR2020IP-00156, Codice Progetto: PRJ-0695. AM, BK, FD, FR, JK, JN, JZ, KN, MG, MR, MS, RB funded by Ministry of Science and Higher Education of Poland with grant 28/WFSN/2021 to the University of Warsaw. BRe, CPe, JLAz funded by Ministerio de Sanidad/ISCIII. BT, PG funded by PERISCOPE European H2020 project, contract number 101016233. CP, DL, EA, MC, SA funded by European Commission - Directorate-General for Communications Networks, Content and Technology through the contract LC-01485746, and Ministerio de Ciencia, Innovacion y Universidades and FEDER, with the project PGC2018-095456-B-I00. DE., MGu funded by Spanish Ministry of Health / REACT-UE (FEDER). DO, GF, IMi, LC funded by Laboratory Directed Research and Development program of Los Alamos National Laboratory (LANL) under project number 20200700ER. DS, ELR, GG, NGR, NW, YW funded by National Institutes of General Medical Sciences (R35GM119582; the content is solely the responsibility of the authors and does not necessarily represent the official views of NIGMS or the National Institutes of Health). FB, FP funded by InPresa, Lombardy Region, Italy. HG, KS funded by European Centre for Disease Prevention and Control. IV funded by Agencia de Qualitat i Avaluacio Sanitaries de Catalunya (AQuAS) through contract 2021-021OE. JDe, SMo, VP funded by Netzwerk Universitatsmedizin (NUM) project egePan (01KX2021). JPB, SH, TH funded by Federal Ministry of Education and Research (BMBF; grant 05M18SIA). KH, MSc, YKh funded by Project SaxoCOV, funded by the German Free State of Saxony. Presentation of data, model results and simulations also funded by the NFDI4Health Task Force COVID-19 (https://www.nfdi4health.de/task-force-covid-19-2) within the framework of a DFG-project (LO-342/17-1). LP, VE funded by Mathematical and Statistical modelling project (MUNI/A/1615/2020), Online platform for real-time monitoring, analysis and management of epidemic situations (MUNI/11/02202001/2020); VE also supported by RECETOX research infrastructure (Ministry of Education, Youth and Sports of the Czech Republic: LM2018121), the CETOCOEN EXCELLENCE (CZ.02.1.01/0.0/0.0/17-043/0009632), RECETOX RI project (CZ.02.1.01/0.0/0.0/16-013/0001761). NIB funded by Health Protection Research Unit (grant code NIHR200908). SAb, SF funded by Wellcome Trust (210758/Z/18/Z).

## Editor's evaluation

This large-scale collaborative study is a timely contribution that will be of interest to researchers working in the fields of infectious disease forecasting and epidemic control. This paper provides a comprehensive evaluation of the predictive skills of real-time COVID-19 forecasting models in

Europe. The conclusions of the paper are well supported by the data and are consistent with findings from studies in other countries.

## Introduction

Epidemiological forecasts make quantitative statements about a disease outcome in the near future. Forecasting targets can include measures of prevalent or incident disease and its severity, for some population over a specified time horizon. Researchers, policy makers, and the general public have used such forecasts to understand and respond to the global outbreaks of COVID-19 (*Van Basshuysen et al., 2021*; *CDC, 2020*; *European Centre for Disease Prevention and Control, 2021c*). At the same time, forecasters use a variety of methods and models for creating and publishing forecasts, varying in both defining the forecast outcome and in reporting the probability distribution of outcomes (*Zelner et al., 2021*; *James et al., 2021*).

Within Europe, comparing forecasts across both models and countries can support a range of national policy needs simultaneously. European public health professionals operate across national, regional, and continental scales, with strong existing policy networks in addition to rich patterns of cross-border migration influencing epidemic dynamics. A majority of European countries also cooperate in setting policy with inter-governmental European bodies such as the European Centre for Disease Prevention and Control (ECDC). In this case, a consistent approach to forecasting across the continent as a whole can support accurately informing cross-European monitoring, analysis, and guidance (*European Centre for Disease Prevention and Control, 2021c*). At a regional level, multi-country forecasts can support a better understanding of the impact of regional migration networks. Meanwhile, where there is limited capacity for infectious disease forecasting at a national level, forecasters generating multi-country results can provide an otherwise-unavailable opportunity for forecasts to inform national situational awareness. Some independent forecasting models have sought to address this by producing multi-country results (*Aguas et al., 2020*; *Adib et al., 2021*; *Agosto and Giudici, 2020*; *Agosto et al., 2021*).

Variation in forecast methods and presentation makes it difficult to compare predictive performance between forecast models, and from there to derive objective arguments for using one forecast over another. This confounds the selection of a single representative forecast and reduces the reliability of the evidence base for decisions based on forecasts. A 'forecast hub' is a centralised effort to improve the transparency and usefulness of forecasts, by standardising and collating the work of many independent teams producing forecasts (*Reich et al., 2019a*). A hub sets a commonly agreed-upon structure for forecast targets, such as type of disease event, spatio-temporal units, or the set of quantiles of the probability distribution to include from probabilistic forecasts. For instance, a hub may collect predictions of the total number of cases reported in a given country for each day in the next 2 weeks. Forecasters can adopt this format and contribute forecasts for centralised storage in the public domain.

This shared infrastructure allows forecasts produced from diverse teams and methods to be visualised and quantitatively compared on a like-for-like basis, which can strengthen public and policy use of disease forecasts. The underlying approach to creating a forecast hub was pioneered in climate modelling and adapted for collaborative epidemiological forecasts of dengue (*Johansson et al., 2019*) and influenza in the USA (*Reich et al., 2019a*; *Reich et al., 2019b*). This infrastructure was adapted for forecasts of short-term COVID-19 cases and deaths in the US (*Cramer et al., 2021a*; *Ray et al., 2020*), prompting similar efforts in some European countries (*Bracher et al., 2021c*; *Funk et al., 2020*; *Bicher et al., 2020*).

Standardising forecasts allows for combining multiple forecasts into a single ensemble with the potential for an improved predictive performance. Evidence from previous efforts in multi-model infectious disease forecasting suggests that forecasts from an ensemble of models can be consistently high performing compared to any one of the component models (*Johansson et al., 2019*; *Reich et al., 2019b*; *Viboud et al., 2018*). Elsewhere, weather forecasting has a long-standing use of building ensembles of models using diverse methods with standardised data and formatting in order to improve performance (*Buizza, 2019*; *Moran et al., 2016*).

The European COVID-19 Forecast Hub (*European Covid-19 Forecast Hub, 2023d*) is a project to collate short-term forecasts of COVID-19 across 32 countries in the European region. The Hub is

funded and supported by the ECDC, with the primary aim to provide reliable information about the near-term epidemiology of the COVID-19 pandemic to the research and policy communities and the general public (*European Centre for Disease Prevention and Control, 2021c*). Second, the Hub aims to create infrastructure for storing and analysing epidemiological forecasts made in real time by diverse research teams and methods across Europe. Third, the Hub aims to maintain a community of infectious disease modellers underpinned by open science principles.

We started formally collating and combining contributions to the European Forecast Hub in March 2021. Here, we investigate the predictive performance of an ensemble of all forecasts contributed to the Hub in real time each week, as well as the performance of variations of ensemble methods created retrospectively.

## Materials and methods

We developed infrastructure to host and analyse prospective forecasts of COVID-19 cases and deaths. The infrastructure is compatible with equivalent research software from the US (*Cramer et al., 2021c*; *Wang et al., 2021*) and German and Polish COVID-19 (*Bracher et al., 2020*) Forecast Hubs, and easy to replicate for new forecasting collaborations.

### Forecast targets and models

We sought forecasts for the incidence of COVID-19 as the total reported number of cases and deaths per week. We considered forecasts for 32 countries in Europe, including all countries of the European Union, European Free Trade Area, and the United Kingdom. We compared forecasts against observed data reported for each country by Johns Hopkins University (JHU, *Dong et al., 2020*). JHU data sources included a mix of national and aggregated subnational data. We aggregated incidence over the Morbidity and Mortality Weekly Report (MMWR) epidemiological week definition of Sunday through Saturday.

Teams could express their uncertainty around any single forecast target by submitting predictions for up to 23 quantiles (from 0.01 to 0.99) of the predictive probability distribution. Teams could also submit a single point forecast. At the first submission, we asked teams to add a pre-specified set of metadata briefly describing the forecasting team and methods (provided online and in supplementary information). No restrictions were placed on who could submit forecasts. To increase participation, we actively contacted known forecasting teams across Europe and the US and advertised among the ECDC network. Teams submitted a broad spectrum of model types, ranging from mechanistic to empirical models, agent-based and statistical models, and ensembles of multiple quantitative or qualitative models (described at *European Covid-19 Forecast Hub, 2023a*). We maintain a full project specification with a detailed submissions protocol (*European Covid-19 Forecast Hub, 2023c*).

We collected forecasts submitted weekly in real time over the 52-week period from 08 March 2021 to 07 March 2022. Teams submitted at latest 2 days after the complete dataset for the latest forecasting week became available each Sunday. We implemented an automated validation programme to check that each new forecast conformed to standardised formatting. Forecast validation ensured a monotonic increase of predictions with each increasing quantile, integer-valued non-negative counts of predicted cases, as well as consistent date and location definitions.

Each week we used all available valid forecasts to create a weekly real-time ensemble model (referred to as 'the ensemble' from here on), for each of the 256 possible forecast targets: incident cases and deaths in 32 locations over the following one through 4 weeks. The ensemble method was an unweighted average of all models' forecast values, at each predictive quantile for a given location, target, and horizon. From 08 March 2021, we used the arithmetic mean. However we noticed that including highly anomalous forecasts in a mean ensemble produced extremely wide uncertainty. To mitigate this, from 26th July 2021 onwards the ensemble instead used a median of all predictive quantiles.

We created an open and publicly accessible interface to the forecasts and ensemble, including an online visualisation tool allowing viewers to see past data and interact with one or multiple forecasts for each country and target for up to 4 weeks' horizon (*European Covid-19 Forecast Hub, 2023b*). All forecasts, metadata, and evaluations are freely available and held on Github (*European Covid-19 Forecast Hub, 2023d*) (archived in real-time at *Sherratt, 2022*), and Zoltar, a platform for hosting

epidemiological forecasts (*EpiForecasts, 2021*; *Reich et al., 2021*). In the codebase for this study (*covid19-forecast-hub-europe, 2022*) we provide a simple method and instructions for downloading and preparing these data for analysis using R. We encourage other researchers to freely use and adapt this to support their own analyses.

## Forecast evaluation

In this study, we focused only on the comparative performance of forecasting models relative to each other. Performance in absolute terms is available on the Hub website (*European Covid-19 Forecast Hub, 2023b*). For each model, we assessed calibration and overall predictive performance. We evaluated all previous forecasts against actual observed values for each model, stratified by the forecast horizon, location, and target. We calculated scores using the *scoringutils* R package (*Bosse et al., 2023*). We removed any forecast surrounding (both the week of, and the first week after) a strongly anomalous data point. We defined anomalous as where any subsequent data release revised that data point by over 5%.

To investigate calibration, we assessed coverage as the correspondence between the forecast probability of an event and the observed frequency of that event. This usage follows previous work in epidemic forecasting (*Bracher et al., 2021a*), and is related to the concept of reliability for binary forecasts. We established the accuracy of each model's prediction boundaries as the coverage of the predictive intervals. We calculated coverage at a given interval level $k$, where $k \in [0, 1]$, as the proportion $p$ of observations that fell within the corresponding central predictive intervals across locations and forecast dates. A perfectly calibrated model would have $p = k$ at all 11 levels (corresponding to 22 quantiles excluding the median). An underconfident model at level $k$ would have $p > k$, i.e. more observations fall within a given interval than expected. In contrast, an overconfident model at level $k$ would have $p < k$, i.e. fewer observations fall within a given interval than expected. We here focus on coverage at the $k = 0.5$ and $k = 0.95$ levels.

We also assessed the overall predictive performance of weekly forecasts using the Weighted Interval Score~(WIS) across all available quantiles. The WIS represents a parsimonious approach to scoring forecasts based on uncertainty represented as forecast values across a set of quantiles (*Bracher et al., 2021a*), and is a strictly proper scoring rule, that is, it is optimal for predictions that come from the data-generating model. As a consequence, the WIS encourages forecasters to report predictions representing their true belief about the future (*Gneiting and Raftery, 2007*). Each forecast for a given location and date is scored based on an observed count of weekly incidence, the median of the predictive distribution and the predictive upper and lower quantiles corresponding to the central predictive interval level.

Not all models provided forecasts for all locations and dates, and we needed to compare predictive performance in the face of various levels of missingness across each forecast target. Therefore we calculated a relative WIS. This is a measure of forecast performance which takes into account that different teams may not cover the same set of forecast targets (i.e. weeks and locations). The relative WIS is computed using a *pairwise comparison tournament* where for each pair of models a mean score ratio is computed based on the set of shared targets. The relative WIS of a model with respect to another model is then the ratio of their respective geometric mean of the mean score ratios, such that smaller values indicate better performance.

We scaled the relative WIS of each model with the relative WIS of a baseline model, for each forecast target, location, date, and horizon. The baseline model assumes case or death counts stay the same as the latest data point over all future horizons, with expanding uncertainty, described previously in *Cramer et al., 2021b*. In this study, we report the relative WIS of each model with respect to the baseline model.

## Retrospective ensemble methods

We retrospectively explored alternative methods for combining forecasts for each target at each week. A natural way to combine probability distributions available in the quantile format *Genest, 1992* used here is

$$F^{-1}(\alpha) = \sum_{i=1}^{n} w_i F_i^{-1}(\alpha),$$

Where $F_1 \ldots F_n$ are the cumulative distribution functions of the individual probability distributions (in our case, the predictive distributions of each forecast model $i$ contributed to the hub), $w_i$ are a set of weights in $[0, 1]$; and $\alpha$ are the quantile levels, such that following notation introduced in **Genest, 1992**,

$$F^{-1}(\alpha) = \inf\{t : F_i(t) \geq \alpha\}.$$

Different ensemble choices then mainly translate to the choice of weights $w_i$. An arithmetic mean ensemble uses weights at $w_i = 1/n$, where all weights are equal and sum up to 1.

Alternatively, we can choose a set of weights to apply to forecasts before they are combined. Numerous options exist for choosing these weights with the aim to maximise predictive performance, including choosing weights to reflect each forecast's past performance (thereby moving from an untrained to a trained ensemble). A straightforward choice is so-called inverse score weighting. In this case, the weights are calculated as

$$w_i = \frac{1}{S_i},$$

where $S_i$ reflects the forecasting skill calculated as the relative WIS of forecaster $i$, calculated over all available model data, and normalised so that weights sum to 1. This method of weighting was found in the US to outperform unweighted scores during some time periods (**Taylor and Taylor, 2023**) but this was not confirmed in a similar study in Germany and Poland (**Bracher et al., 2021c**).

When constructing ensembles from quantile means, a single outlier can have an oversized effect on the ensemble forecast. Previous research has found that a median ensemble, replacing the arithmetic mean of each quantile with a median of the same values, yields competitive performance while

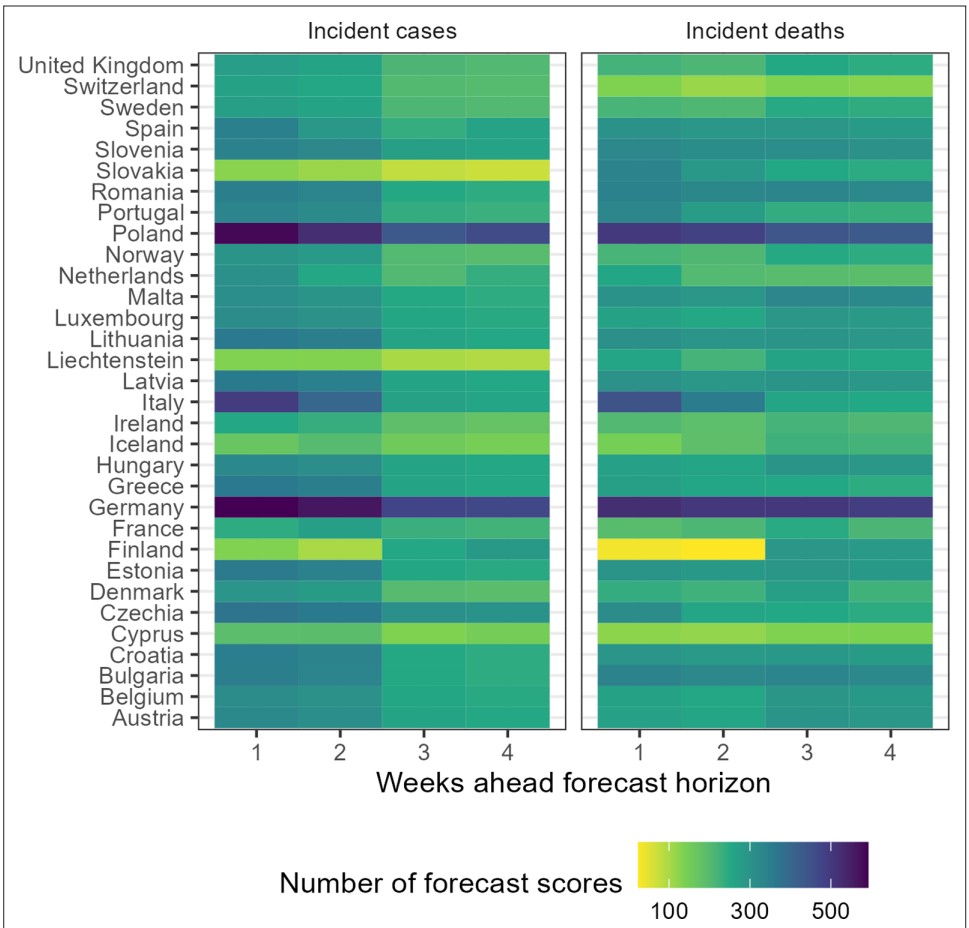

**Figure 1.** Total number of forecasts included in evaluation, by target location, week ahead horizon, and variable.

maintaining robustness to outlying forecasts (*Ray et al., 2022*). Building on this, we also created weighted median ensembles using the weights described above and a Harrel-Davis quantile estimator with a beta function to approximate the weighted percentiles (*Harrell and Davis, 1982*). We then compared the performance of unweighted and inverse relative WIS weighted mean and median ensembles, comparing the ratio of interval scores between each ensemble model relative to the baseline model.

## Results

For 32 European countries, we collected, visualised, and made available online weekly COVID-19 forecasts and observed data (*Sherratt, 2022*). Over the whole study period, we collected forecasts from 48 unique models. Modellers created forecasts choosing from a set of 32 possible locations, four time horizons, and two variables, and modellers variously joined and left the Hub over time. This meant the number of models contributing to the Hub varied over time and by forecasting target. Using all models and the ensemble, we created 2139 forecasting scores, where each score summarises a unique combination of forecasting model, variable, country, and week ahead horizon (*Figure 1*).

Of the total 48 models, we received the most forecasts for Germany, with 29 unique models submitting 1-week case forecasts, while only 12 models ever submitted 4-week case or death forecasts for Liechtenstein. Modelling teams also differed in how they expressed uncertainty. Only three models provided point forecasts with no estimate of uncertainty around their predictions, while 41 models provided the full set of 23 probabilistic quantiles across the predictive distribution for each target.

In this evaluation we included 29 models in comparison to the ensemble forecast (*Figure 1*). We have included metadata provided by modellers in the supplement and online (*Sherratt, 2022*). In this evaluation, at most 15 models contributed forecasts for cases in Germany at the 1 week horizon, with an accumulated 592 forecast scores for that single target over the study period. In contrast, deaths in Finland at the 2 week horizon saw the smallest number of forecasts, with only 6 independent models contributing 24 forecast scores at any time over the 52-week period. Of the 29 models included in this evaluation, 5 models provided less than the full set of 23 quantiles, and were excluded when creating the ensemble. No ensemble forecast was composed of less than 3 independent models.

We visually compared the absolute performance of forecasts in predicting numbers of incident cases and deaths. We observed that forecasts predicted well in times of stable epidemic behaviour,

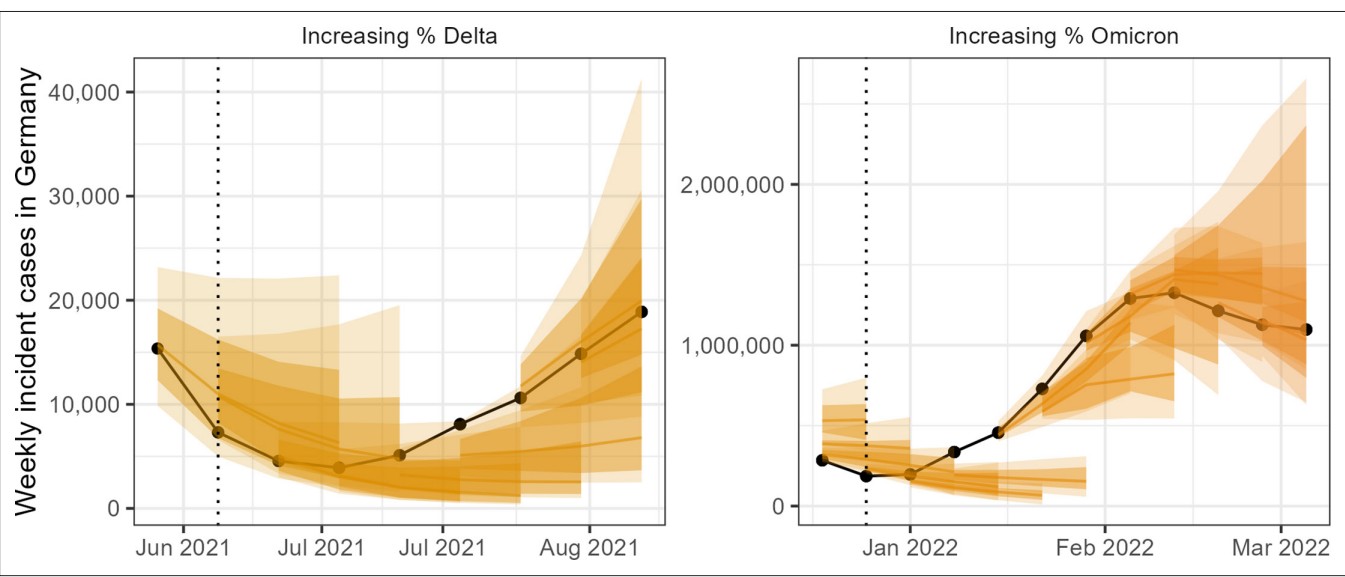

**Figure 2.** Ensemble forecasts of weekly incident cases in Germany over periods of increasing SARS-CoV-2 variants Delta (B.1.617.2, left) and Omicron (B.1.1.529, right). Black indicates observed data. Coloured ribbons represent each weekly forecast of 1–4 weeks ahead (showing median, 50%, and 90% probability). For each variant, forecasts are shown over an x-axis bounded by the earliest dates at which 5% and 99% of sequenced cases were identified as the respective variant of concern, while vertical dotted lines indicate the approximate date that the variant reached dominance (>50% sequenced cases).

while struggling to accurately predict at longer horizons around inflection points, for example during rapid changes in population-level behaviour or surveillance. Forecast models varied widely in their ability to predict and account for the introduction of new variants, giving the ensemble forecast over these periods a high level of uncertainty. An example of weekly forecasts from the ensemble model is shown in *Figure 2*.

In relative terms, the ensemble of all models performed well compared to both its component models and the baseline. By relative WIS scaled against a baseline of 1 (where a score <1 indicates outperforming the baseline), the median score of forecasts from the Hub ensemble model was 0.71, within an interquartile range of 0.61 at 25% probability to 0.88 at 75% probability. Meanwhile the median score of forecasts across all participating models (excluding the Hub ensemble) was 1.04 (IQR 0.82–1.36).

Across all horizons and locations, the ensemble performed better on scaled relative WIS than 83% of forecast scores when forecasting cases (with a total N=886 from 23 unique models), and 91% of scores for forecasts of incident deaths (N=763 scores from 20 models). We also saw high performance from the ensemble when evaluating against all models including those who did not submit the full set of probabilistic quantile predictions (80% for cases with N=1006 scores from 28 models, and 88% for deaths, N=877 scores from 24 models).

The performance of individual and ensemble forecasts varied by length of the forecast horizon (*Figure 3*). At each horizon, the typical performance of the ensemble outperformed both the baseline model and the aggregated scores of all its component models, although we saw wide variation between individual models in performance across horizons. Both individual models and the ensemble saw a trend of worsening performance at longer horizons when forecasting cases with the median scaled relative WIS of the ensemble across locations worsened from 0.62 for 1-week ahead forecasts to 0.9 when forecasting 4 weeks ahead. Performance for forecasts of deaths was more stable over one through 4 weeks, with median ensemble performance moving from 0.69 to 0.76 across the 4-week horizons.

We observed similar trends in performance across horizon when considering how well the ensemble was calibrated with respect to the observed data. At 1 week ahead the case ensemble was well calibrated (ca. 50% and 95% nominal coverage at the 50% and 95% levels, respectively). This did not hold at longer forecast horizons as the case forecasts became increasingly over-confident. Meanwhile, the ensemble of death forecasts was well calibrated at the 95% level across all horizons, and the calibration of death forecasts at the 50% level improved with lengthening horizons compared to being underconfident at shorter horizons.

The ensemble also performed consistently well in comparison to individual models when forecasting across countries (*Figure 4*). In total, across 32 countries forecasting for 1 through 4 weeks, when forecasting cases the ensemble outperformed 75% of component models in 22 countries, and outperformed all available models in 3 countries. When forecasting deaths, the ensemble outperformed 75% and 100% of models in 30 and 8 countries, respectively. Considering only the the 2-week horizon shown in *Figure 4*, the ensemble of case forecasts outperformed 75% models in 25 countries and all models in only 12 countries. At the 2-week horizon for forecasts of deaths, the ensemble outperformed 75% and 100% of its component models in 30 and 26 countries, respectively.

We considered alternative methods for creating ensembles from the participating forecasts, using either a mean or median to combine either weighted or unweighted forecasts. We evaluated each alternative ensemble model against the baseline model, taking the mean score ratio across all targets (*Table 1*). Across locations we observed that the median outperformed the mean across all one through 4 week horizons and both cases and death targets, for all but cases at the 1 week horizon. This held regardless of whether the component forecasts were weighted or unweighted by their individual past performance. Between methods of combination, weighting made little difference to the performance of the median ensemble, but appeared to improve performance of a mean ensemble in forecasting deaths.

## Discussion

We collated 12 months of forecasts of COVID-19 cases and deaths across 32 countries in Europe, collecting from multiple independent teams and using a principled approach to standardising both forecast targets and the predictive distribution of forecasts. We combined these into an ensemble

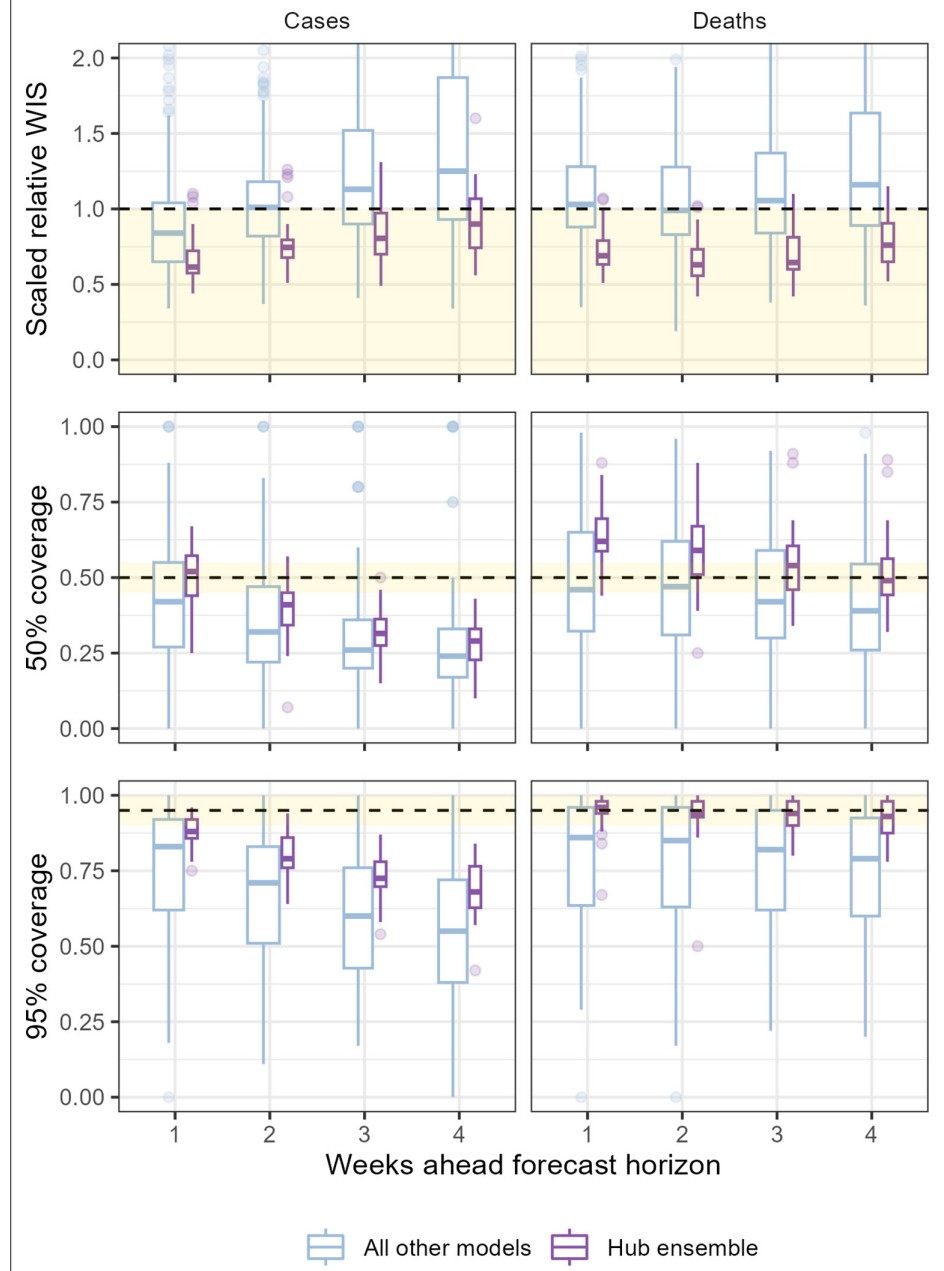

**Figure 3.** Performance of short-term forecasts aggregated across all individually submitted models and the Hub ensemble, by horizon, forecasting cases (left) and deaths (right). Performance measured by relative weighted interval score scaled against a baseline (dotted line, 1), and coverage of uncertainty at the 50% and 95% levels. Boxplot, with width proportional to number of observations, show interquartile ranges with outlying scores as faded points. The target range for each set of scores is shaded in yellow.

forecast and compared the relative performance of forecasts between models, finding that the ensemble forecasts outperformed most individual models across all countries and horizons over time.

Across all models we observed that forecasting changes in trend in real time was particularly challenging. Our study period included multiple fundamental changes in viral-, individual-, and population-level factors driving the transmission of COVID-19 across Europe. In early 2021, the introduction of vaccination started to change population-level associations between infections, cases, and deaths (***European Centre for Disease Prevention and Control, 2021b***), while the Delta variant emerged and became dominant (***European Centre for Disease Prevention and Control, 2021a***). Similarly from

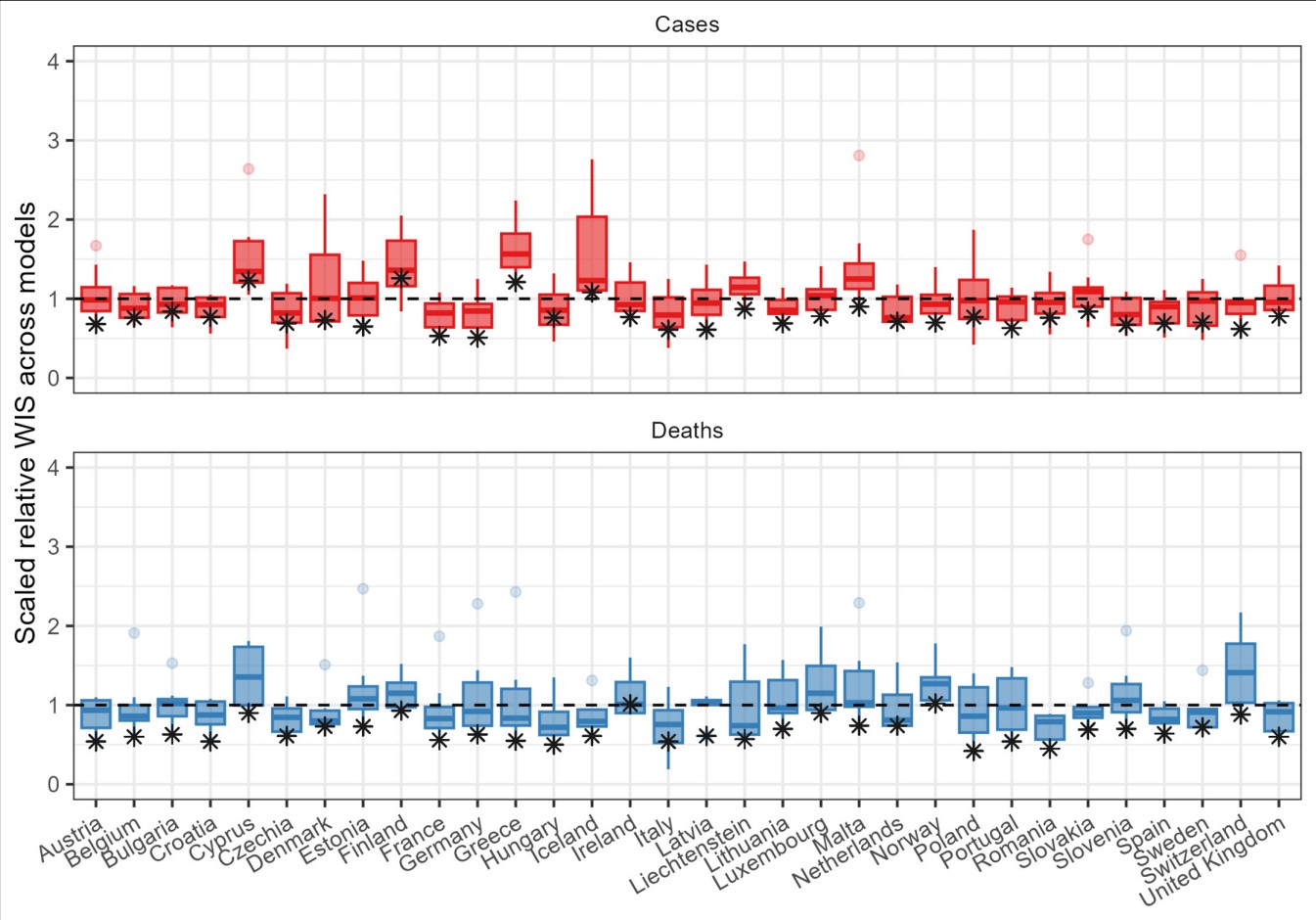

**Figure 4.** Performance of short-term forecasts across models and median ensemble (asterisk), by country, forecasting cases (top) and deaths (bottom) for 2-week ahead forecasts, according to the relative weighted interval score. Boxplots show interquartile ranges, with outliers as faded points, and the ensemble model performance is marked by an asterisk. y-axis is cut-off to an upper bound of 4 for readability.

**Table 1.** Predictive performance of main ensembles, as measured by the mean ratio of interval scores against the baseline ensemble.

| Horizon | Weighted mean | Weighted median | Unweighted mean | Unweighted median |
|---------|---------------|-----------------|-----------------|-------------------|
| Cases | | | | |
| 1 week | 0.63 | 0.64 | 0.61 | 0.64 |
| 2 weeks | 0.72 | 0.71 | 0.69 | 0.69 |
| 3 weeks | 0.82 | 0.76 | 0.82 | 0.72 |
| 4 weeks | 1.07 | 0.86 | 1.12 | 0.78 |
| Deaths | | | | |
| 1 week | 0.65 | 0.61 | 1.81 | 0.61 |
| 2 weeks | 0.58 | 0.54 | 1.29 | 0.54 |
| 3 weeks | 0.64 | 0.57 | 1.17 | 0.53 |
| 4 weeks | 0.82 | 0.67 | 0.84 | 0.62 |

late 2021 we saw the interaction of individually waning immunity during the emergence and global spread of the Omicron variant (*European Centre for Disease Prevention and Control, 2022b*). Neither the extent nor timing of these factors were uniform across European countries covered by the Forecast Hub (*European Centre for Disease Prevention and Control, 2023*). This meant that the performance of any single forecasting model depended partly on the ability, speed, and precision with which it could adapt to new conditions for each forecast target.

We observed a contrast between a more stable performance of forecasting deaths further into the future compared to forecasts of cases. Previous work has found rapidly declining performance for case forecasts with increasing horizon (*Cramer et al., 2021b*; *Castro et al., 2020*), while death forecasts can perform well with up to 6 weeks lead time (*Friedman et al., 2021*). We can link this to the specific epidemic dynamics in this study.

First, COVID-19 has a typical serial interval of less than a week (*Alene et al., 2021*). This implies that case forecasts of more than 2 weeks only remain valid if rates of both transmission and detection remain stable over the entire forecast horizon. In contrast, we saw rapid changes in epidemic dynamics across many countries in Europe over our study period, impacting the longer term case forecasts.

Second, we can interpret the higher reliability of death forecasts as due to the different lengths and distributions of time lags from infection to case and death reporting (*Jin, 2021*). For example, a spike in infections may be matched by a consistently sharp increase in case reporting, but a longer tailed distribution of the subsequent increase in death reports. This creates a lower magnitude of fluctuation in the time-series of deaths compared to that of cases. Similarly, surveillance data for death reporting is substantially more consistent, with fewer errors and retrospective corrections, than case reporting (*Català et al., 2021*).

Third, we also note that the performance of trend-based forecasts may have benefited from the slower changes to trends in incident deaths caused by gradually increasing vaccination rates. These features allow forecasters to incorporate the effect of changes in transmission more easily when forecasting deaths, compared to cases.

We found the ensemble in this study continued to outperform both other models and the baseline at up to 4 weeks ahead. Our results support previous findings that ensemble forecasts are the best or nearly the best performing models with respect to absolute predictive performance and appropriate coverage of uncertainty (*Funk et al., 2020*; *Viboud et al., 2018*; *Cramer et al., 2021b*). While the ensemble was consistently high performing, it was not strictly dominant across all forecast targets, reflecting findings from previous comparable studies of COVID-19 forecasts (*Bracher et al., 2021c*; *Brooks, 2020*). Our finding suggests the usefulness of an ensemble as a robust summary when forecasting across many spatio-temporal targets, without replacing the importance of communicating the full range of model predictions.

When exploring variations in ensemble methods, we found that the choice of median over means yielded the most consistent improvement in predictive performance, regardless of the method of weighting. Other work has supported the importance of the median in providing a stable forecast that better accounts for outlier forecasts than the mean (*Brooks, 2020*), although this finding may be dependent on the quality of the individual forecast submissions. In contrast, weighing models by past performance did not result in any consistent improvement in performance. This is in line with existing mixed evidence for any optimal ensemble method for combining short term probabilistic infectious disease forecasts. Many methods of combination have performed competitively in analyses of forecasts for COVID-19 in the US, including the simple mean and weighted approaches outperforming unweighted or median methods (*Taylor and Taylor, 2023*). This contrasts with later analyses finding weighted methods to give similar performance to a median average (*Ray et al., 2020*; *Brooks, 2020*). We can partly explain this inconsistency if performance of each method depends on the outcome being predicted (cases, deaths), its count (incident, cumulative) and absolute level, the changing disease dynamics, and the varying quality and quantity of forecasting teams over time.

We note several limitations in our approach to assessing the relative performance of an ensemble among forecast models. While we have described differences in model scores, we have not used any formal statistical test for comparing forecast scores, such as the Diebold-Mariano test (*Diebold and Mariano, 1995*), recognising that it is unclear how this is best achieved across many models. Our results are the outcome of evaluating forecasts against a specific performance metric and baseline, where multiple options for evaluation exist and the choice reflects the aim of the evaluation process.

Further, our choice of baseline model affects the given performance scores in absolute terms, and more generally the choice of appropriate baseline for epidemic forecast models is not obvious when assessing infectious disease forecasts. The model used here is supported by previous work (*Cramer et al., 2021b*), yet previous evaluation in a similar context has suggested that choice of baseline affects relative performance in general (*Bracher et al., 2021b*), and future research should be done on the best choices of baseline models in the context of infectious disease epidemics.

Our assessment of forecast performance may further have been inaccurate due to limitations in the observed data against which we evaluated forecasts. We sourced data from a globally aggregated database to maintain compatibility across 32 countries (*Dong et al., 2020*). However, this made it difficult to identify the origin of lags and inconsistencies between national data streams, and to what extent these could bias forecasts for different targets. In particular, we saw some real time data revised retrospectively, introducing bias in either direction where the data used to create forecasts was not the same as that used to evaluate it. We attempted to mitigate this by using an automated process for determining data revisions, and excluding forecasts made at a time of missing, unreliable, or heavily revised data. We also recognise that evaluating forecasts against updated data is a valid alternative approach used elsewhere (*Cramer et al., 2021b*). More generally it is unclear if the expectation of observation revisions should be a feature built into forecasts. Further research is needed to understand the perspective of end-users of forecasts in order to assess this.

The focus of this study was describing and summarising an ensemble of many models. We note that we have little insight into the individual methods and wide variety of assumptions that modellers used. While we asked modellers to provide a short description of their methods, we did not create a rigorous framework for this, and we did not document whether modellers changed the methods for a particular submitted model over time. Both the content of and variation in modelling methods and assumptions are likely to be critical to explaining performance, rather than describing or summarising it. Exploring modellers' methods and relating this to forecast performance will be an important area of future work.

In an emergency setting, access to visualised forecasts and underlying data is useful for researchers, policymakers, and the public (*CDC, 2020*). Previous European multi-country efforts to forecast COVID-19 have included only single models adapted to country-specific parameters (*Aguas et al., 2020*; *Adib et al., 2021*; *Agosto et al., 2021*).

The European Forecasting Hub acted as a unique tool for creating an open-access, cross-country modelling network, and connecting this to public health policy across Europe. By opening participation to many modelling teams and with international high participation, we were able to create robust ensemble forecasts across Europe. This also allows comparison across forecasts built with different interpretations of current data, on a like for like scale in real time. The European Hub has supported policy outputs at an international, regional, and national level, including Hub forecasts cited weekly in ECDC Communicable Disease Threats Reports (*European Centre for Disease Prevention and Control, 2022a*).

For forecast producers, an easily accessible comparison between results from different methods can highlight individual strengths and weaknesses and help prioritise new areas of work. Collating time-stamped predictions ensures that we can test true out-of-sample performance of models and avoid retrospective claims of performance. Testing the limits of forecasting ability with these comparisons forms an important part of communicating any model-based prediction to decision makers. For example, the weekly ECDC Communicable Disease Threats reports include the specific results of this work by qualitatively highlighting the greater uncertainty around case forecasts compared to death forecasts.

This study raises many further questions which could inform epidemic forecast modellers and users. The dataset created by the European Forecast Hub is an openly accessible, standardised, and extensively documented catalogue of real time forecasting work from a range of teams and models across Europe (*European Covid-19 Forecast Hub, 2023b*), and we recommend its use for further research on forecast performance. In the code developed for this study, we provide a worked example of downloading and using both the forecasts and their evaluation scores (*covid19-forecast-hub-europe, 2022*).

Future work could explore the impact on forecast models of changing epidemiology at a broad spatial scale by combining analyses of trends and turning points in cases and deaths with forecast

performance, or extending to include data on vaccination, variant, or policy changes over time. There is also much scope for future research into methods for combining forecasts to improve performance of an ensemble. This includes altering the inclusion criteria of forecast models based on different thresholds of past performance, excluding or including only forecasts that predict the lowest and highest values (trimming) (*Taylor and Taylor, 2023*), or using alternative weighting methods such as quantile regression averaging (*Funk et al., 2020*). Exploring these questions would add to our understanding of real time performance, supporting and improving future forecasting efforts.

We see additional scope to adapt the Hub format to the changing COVID-19 situation across Europe. We have extended the Forecast Hub infrastructure to include short term forecasts for hospitalisations with COVID-19, which is a challenging task due to limited data across the locations covered by the hub. As the policy focus shifts from immediate response to anticipating changes brought by vaccinations or the geographic spread of new variants (*European Centre for Disease Prevention and Control, 2023*), we are also separately investigating models for longer term scenarios in addition to the short term forecasts in a similar framework to existing scenario modelling work in the US (*Borchering et al., 2021*).

In conclusion, we have shown that during a rapidly evolving epidemic spreading through multiple populations, an ensemble forecast performed highly consistently across a large matrix of forecast targets, typically outperforming the majority of its separate component models and a naive baseline model. In addition, we have linked issues with the predictability of short-term case forecasts to underlying COVID-19 epidemiology, and shown that ensemble methods based on past model performance were unable to reliably improve forecast performance. Our work constitutes a step towards both unifying COVID-19 forecasts and improving our understanding of them.

## Additional information

### Competing interests

Prasith Baccam, Heidi Gurung, Steven Stage, Bradley Suchoski: Affiliated with IEM, Inc. The author has no financial interests to declare. The other authors declare that no competing interests exist.

### Funding

| Funder | Grant reference number | Author |
|---|---|---|
| Netzwerk Universitätsmedizin | Project egePan 01KX2021 | Jonas Dehning Sebastian Mohr Viola Priesemann |
| FISR | SMIGE - Modelli statistici inferenziali per governare l'epidemia, FISR 2020 - Covid-19 I Fase, FISR2020IP_00156, Codice Progetto - PRJ-0695 | Antonello Maruotti Gianfranco Lovison Alessio Farcomeni |
| Agència de Qualitat i Avaluació Sanitàries de Catalunya | Contract 2021_021OE | Inmaculada Villanueva |
| European Centre for Disease Prevention and Control | | Katharine Sherratt |
| European Commission | Communications Networks Content and Technology LC-01485746, Ministerio CIU/FEDER PGC2018-095456-B-I00 | Sergio Alonso Enric Alvarez Daniel Lopez Clara Prats |
| Bundesministerium für Bildung und Forschung | 05M18SIA | Stefan Heyder Thomas Hotz Jan Pablo Burgard |
| Health Protection Research Unit | NIHR200908 | Nikos I Bosse |
| InPresa | Lombardy Region Italy | Fulvia Pennoni Francesco Bartolucci |

| Funder | Grant reference number | Author |
| --- | --- | --- |
| Los Alamos National Laboratory | | Lauren Castro |
| MUNI | Mathematical and Statistical modelling project (MUNI/ A/1615/2020),MUNI/11/02202001/2020 | Veronika Eclerova Lenka Pribylova |
| Ministerio de Sanidad | | Cesar Perez Alvarez |
| Ministry of Science and Higher Education of Poland | 28/WFSN/2021 | Rafal P Bartczuk |
| National Institute of General Medical Sciences | R35GM119582 | Graham Gibson |
| National Institutes of Health | 1R01GM109718 | Lijing Wang |
| Virginia Department of Health | VDH-21-501-0141 | Aniruddha Adiga |
| Virginia Department of Health | VDH-21-501-0143 | Benjamin Hurt |
| Virginia Department of Health | VDH-21-501-0147 | Bryan Lewis |
| Virginia Department of Health | VDH-21-501-0142 | Lijing Wang |
| Virginia Department of Health | VDH-21-501-0148 | Madhav Marathe |
| Virginia Department of Health | VDH-21-501-0145 | Przemyslaw Porebski |
| Virginia Department of Health | VDH-21-501-0146 | Srinivasan Venkatramanan |
| Narodowe Centrum Badań i Rozwoju | INFOSTRATEG-I/0022/2021-00 | Biecek Przemyslaw |
| Horizon 2020 | PERISCOPE 101016233 | Paolo Giudici Barbara Tarantino |
| German Free State of Saxony | LO-342/17-1 | Kirsten Holger Yuri Kheifetz Markus Scholz |
| Spanish Ministry of Health, Social Policy and Equality | REACT-UE (FEDER) | David E Singh |
| Wellcome Trust | 210758/Z/18/Z | Sam Abbott |
| RECETOX Přírodovĕdecké Fakulty Masarykovy Univerzity | LM2018121 | Veronika Eclerova |
| CETOCOEN EXCELLENCEC | CZ.02.1.01/0.0/0.0/17-043/0009632 | Veronika Eclerova |
| RECETOX RI project | CZ.02.1.01/0.0/0.0/16-013/0001761 | Veronika Eclerova |

The funders had no role in study design, data collection and interpretation, or the decision to submit the work for publication. For the purpose of Open Access, the authors have applied a CC BY public copyright license to any Author Accepted Manuscript version arising from this submission.

## Author contributions

Katharine Sherratt, Conceptualization, Data curation, Software, Formal analysis, Investigation, Methodology, Writing - original draft, Writing – review and editing; Hugo Gruson, Software, Writing – review

and editing; Rok Grah, Helen Johnson, Rene Niehus, Bastian Prasse, Frank Sandmann, Funding acquisition, Project administration, Writing – review and editing; Jannik Deuschel, Daniel Wolffram, Graham Gibson, Evan L Ray, Nicholas G Reich, Daniel Sheldon, Yijin Wang, Nutcha Wattanachit, Nikos I Bosse, Johannes Bracher, Software, Methodology, Writing – review and editing; Sam Abbott, Validation, Methodology, Writing – review and editing; Alexander Ullrich, Software, Visualization; Lijing Wang, Jan Trnka, Guillaume Obozinski, Tao Sun, Dorina Thanou, Loic Pottier, Ekaterina Krymova, Jan H Meinke, Maria Vittoria Barbarossa, Neele Leithauser, Jan Mohring, Johanna Schneider, Jaroslaw Wlazlo, Jan Fuhrmann, Berit Lange, Isti Rodiah, Prasith Baccam, Heidi Gurung, Steven Stage, Bradley Suchoski, Jozef Budzinski, Robert Walraven, Inmaculada Villanueva, Vit Tucek, Martin Smid, Milan Zajicek, Cesar Perez Alvarez, Sophie R Meakin, Lauren Castro, Geoffrey Fairchild, Isaac Michaud, Dave Osthus, Pierfrancesco Alaimo Di Loro, Antonello Maruotti, Veronika Eclerova, Andrea Kraus, David Kraus, Lenka Pribylova, Bertsimas Dimitris, Michael Lingzhi Li, Soni Saksham, Jonas Dehning, Sebastian Mohr, Viola Priesemann, Grzegorz Redlarski, Benjamin Bejar, Giovanni Ardenghi, Nicola Parolini, Giovanni Ziarelli, Wolfgang Bock, Stefan Heyder, Thomas Hotz, David E Singh, Miguel Guzman-Merino, Jose L Aznarte, David Morina, Sergio Alonso, Enric Alvarez, Daniel Lopez, Clara Prats, Jan Pablo Burgard, Arne Rodloff, Tom Zimmermann, Alexander Kuhlmann, Janez Zibert, Fulvia Pennoni, Fabio Divino, Marti Catala, Gianfranco Lovison, Paolo Giudici, Barbara Tarantino, Francesco Bartolucci, Giovanna Jona Lasinio, Marco Mingione, Alessio Farcomeni, Ajitesh Srivastava, Pablo Montero-Manso, Aniruddha Adiga, Benjamin Hurt, Bryan Lewis, Madhav Marathe, Przemyslaw Porebski, Srinivasan Venkatramanan, Rafal P Bartczuk, Filip Dreger, Anna Gambin, Krzysztof Gogolewski, Magdalena Gruziel-Slomka, Bartosz Krupa, Antoni Moszyński, Karol Niedzielewski, Jedrzej Nowosielski, Maciej Radwan, Franciszek Rakowski, Marcin Semeniuk, Ewa Szczurek, Jakub Zielinski, Jan Kisielewski, Barbara Pabjan, Kirsten Holger, Yuri Kheifetz, Markus Scholz, Biecek Przemyslaw, Marcin Bodych, Maciej Filinski, Radoslaw Idzikowski, Tyll Krueger, Tomasz Ozanski, Methodology, Writing – review and editing; Borja Reina, Methodology, Writing – review and editing, Conceptualization; Sebastian Funk, Conceptualization, Software, Supervision, Writing – review and editing

## Author ORCIDs

Katharine Sherratt ⓘ http://orcid.org/0000-0003-2049-3423
Daniel Wolffram ⓘ http://orcid.org/0000-0003-0318-3669
Yijin Wang ⓘ http://orcid.org/0000-0003-4438-6366
Jan Trnka ⓘ http://orcid.org/0000-0002-1786-7562
Tao Sun ⓘ http://orcid.org/0000-0001-6357-6726
Johanna Schneider ⓘ http://orcid.org/0000-0002-9330-2838
Jan Fuhrmann ⓘ http://orcid.org/0000-0002-7091-3740
Inmaculada Villanueva ⓘ http://orcid.org/0000-0003-4940-085X
Milan Zajicek ⓘ http://orcid.org/0000-0002-3226-7266
Antonello Maruotti ⓘ http://orcid.org/0000-0001-8377-9950
Veronika Eclerova ⓘ http://orcid.org/0000-0001-8476-7740
Viola Priesemann ⓘ http://orcid.org/0000-0001-8905-5873
Sergio Alonso ⓘ http://orcid.org/0000-0002-3989-8757
Clara Prats ⓘ http://orcid.org/0000-0002-1398-7559
Jan Pablo Burgard ⓘ http://orcid.org/0000-0002-5771-6179
Alessio Farcomeni ⓘ http://orcid.org/0000-0002-7104-5826
Bryan Lewis ⓘ http://orcid.org/0000-0003-0793-6082
Przemyslaw Porebski ⓘ http://orcid.org/0000-0001-8012-5791
Rafal P Bartczuk ⓘ http://orcid.org/0000-0002-0433-7327
Krzysztof Gogolewski ⓘ http://orcid.org/0000-0001-5523-5198
Jakub Zielinski ⓘ http://orcid.org/0000-0001-8935-8137
Sebastian Funk ⓘ http://orcid.org/0000-0002-2842-3406

## Decision letter and Author response

Decision letter https://doi.org/10.7554/eLife.81916.sa1
Author response https://doi.org/10.7554/eLife.81916.sa2

## Additional files

### Supplementary files
• Supplementary file 1. EPIFORGE reporting guidelines Completed checklist following reporting guidelines on epidemic forecasting research.
• Supplementary file 2. Participating team metadata Team metadata for teams participating in the European Forecast Hub and evaluated in this study.
• MDAR checklist

### Data availability
All source data were openly available before the study, originally available at: https://github.com/covid19-forecast-hub-europe/covid19-forecast-hub-europe (copy archived at swh:1:rev:b4d66c495e-07c12d88384506154cf58f08592365). All data and code for this study are openly available on Github: covid19-forecast-hub-europe/euro-hub-ensemble.

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
