## [Editor Report]

This large-scale collaborative study is a timely contribution that will be of interest to researchers working in the fields of infectious disease forecasting and epidemic control. This paper provides a comprehensive evaluation of the predictive skills of real-time COVID-19 forecasting models in Europe. The conclusions of the paper are well supported by the data and are consistent with findings from studies in other countries.

---

## [Decision Letter]

**Decision letter after peer review:**

Thank you for submitting your article "Predictive performance of multi-model ensemble forecasts of COVID-19 across European nations" for consideration by eLife. Your article has been reviewed by 2 peer reviewers, and the evaluation has been overseen by a Reviewing Editor and Neil Ferguson as the Senior Editor. The following individuals involved in the review of your submission have agreed to reveal their identity: Jeffrey L Shaman (Reviewer #1); Sen Pei (Reviewer #2).

Essential revisions:

The primary comment was regarding the novelty and additional insights gained from this work. While both reviewers noted that the methodology was sound, there are other papers reporting very similar findings/work in this setting (and others) but the added value of this work, in particular, was not clear. The authors are encouraged to better articulate the added value of the work.

*Reviewer #1 (Recommendations for the authors):*

I guess my main question is: do we need another report on multi-model 'ensembling'? I'm not sure. This work is more substantive and validated than multi-model scenario efforts (i.e. it is forecasting not scenario play), which are often wildly speculative and in many instances shouldn't be published in high-profile journals (and I've been on a few of those papers).

I will let the editor decide.

A few other comments.

The authors use a flexible submission structure that does not appear to be strictly regularized. They write: 'Teams could express their uncertainty around any single forecast target by submitting predictions for up to 23 quantiles (from 0.01 to 0.99) of the predictive probability distribution. Teams could also submit a single-point forecast.' Were there any issues arising from this? For instance, given that there was flexibility in what was submitted-some only submitting some quantiles or a point prediction, leading to variable missingness across quantiles-are there instances where the average mean or median value does not increase monotonically with quantile?

'Coverage' is an evocative term; in weather, they more typically use 'reliability', defined as the correspondence between the forecast probability of an event and the observed frequency of that event. Consider at least noting that coverage, as defined here, is reliability. Calibration is used to describe reliability, and I note this is used in the text.

The use of relative WIS is nice.

I believe your definition of F^(-1) (α), which is confusing as I reflexively read this as a matrix inverse (perhaps use G(α) instead), is for the supremum (least upper bound), not the infimum (greatest lower bound), i.e. G(α)=sup{t:F_i (t){greater than or equal to}α}. If not, I think the {greater than or equal to} should be {less than or equal to}.

Note that in US flu forecasting, there is an expectation of observation revisions. Forecasts are validated against final revised observations, despite what was available in real time.

*Reviewer #2 (Recommendations for the authors):*

This is a solid and well-written work summarizing the efforts of the European COVID-19 Forecast Hub.

---

## [Author Response]

Comments are grouped by theme.

NoveltyThere are other papers reporting very similar findings/work in this setting (and others) but the added value of this work, in particular, was not clear.I guess my main question is: do we need another report on multi-model 'ensembling'?

We agree with reviewers that our findings add depth rather than breadth to the evidence base for multi-model ensembles in real time infectious disease forecasting. As mentioned by Reviewer #2, this work was unique and unprecedented for European policy makers in spanning multiple countries while aiming to inform continent-wide public health and we believe holds particularly strong value in highlighting the relevance of forecasting at multiple policy-making scales (national, regional, international).

We have added commentary on the specific value of this effort to European policy makers as well as forecast producers in both the background and discussion sections.

Methodological limitations‘Teams could also submit a single-point forecast.' Were there any issues arising from this?

Several teams did submit point forecasts, or forecasts with less than the full set of quantiles (5 of 29 models evaluated here). We have historically reported absolute error for all models in real time but in this paper they are excluded from the evaluations using the interval score, which rely on the full set of quantiles.

The exclusion from the ensemble of forecasts without the full set of quantiles was not clear from the current paper text. We have now updated the text to make this exclusion explicit (Methods section, under “Forecast evaluation”, and re-stated for clarity in the Results).

Note that in US flu forecasting, there is an expectation of observation revisions. Forecasts are validated against final revised observations, despite what was available in real time.

As discussed in the text (Discussion, page 14), we excluded forecasts with revised observations. As we noted: “*More generally it is unclear if the expectation of observation revisions should be a feature built into forecasts. Further research is needed to understand the perspective of end-users of forecasts in order to assess this.”* In the context of this paper we felt the fairest approach to evaluation was to exclude forecasts made for revised observations, while recognising that evaluating forecasts against updated data is a valid approach.

We have added a note on this alternative approach in the discussion.

Definitions and clarifications'Coverage' is an evocative term; in weather, they more typically use 'reliability', defined as the correspondence between the forecast probability of an event and the observed frequency of that event. Consider at least noting that coverage, as defined here, is reliability. Calibration is used to describe reliability, and I note this is used in the text.

Thank you for suggesting this as we had not observed this difference in usage. Here we used coverage in line with previous work discussing interval scores for quantile forecasts.

We have added a note and reference in the Methods section for our use of “coverage”.

I believe your definition of F^(-1) (α), which is confusing as I reflexively read this as a matrix inverse (perhaps use G(α) instead), is for the supremum (least upper bound), not the infimum (greatest lower bound), i.e. G(α)=sup{t:F_i (t){greater than or equal to}α}. If not, I think the {greater than or equal to} should be {less than or equal to}.

We apologise if the notation was perceived as ambiguous. We followed the notation of the original reference cited in the paper^1^ (eq.1.1). As an example, the 0.05 quantile of a distribution with cumulative distribution function F(x) would be the greatest value of t that is a lower bound of the set of all t that fulfil F(t)> = 0.05, which is the definition as written (it would be the supremum if the direction of the inequality was reversed). We could replace it with the minimum here (the lowest t with F(t) > = 0.05) for all practical intents and purposes, but decided to stay with the original notation so that it can be tracked to the given reference.

We have clarified reference to notation in the text.

1. C. Genest, “Vincentization Revisited,” The Annals of Statistics, vol. 20, no. 2, pp. 1137–1142, 1992, Available: https://www.jstor.org/stable/2242003